# An experimental investigation of Lean Six Sigma philosophies in a high-mix low-volume manufacturing environment

Amanda Normand⬥*⬥, T. H. Bradley⬥⬥

Department of Systems Engineering, Walter Scott, Jr. College of Engineering, Colorado State University, Fort Collins, Colorado, United States of America

⬥ These authors contributed equally to this work.
* normanda@colostate.edu

**Data Availability Statement:** Relevant data are within the manuscript and supporting files. Dataset is also publicly available on Dryad (forthcoming

## Abstract

This article experimentally examines methods for implementing the philosophies of Lean Six Sigma (LSS) in a High-Mix Low-Volume (HMLV) manufacturing environment. HMLV environments present unique challenges to LSS paradigms because of the need for extraordinary operational flexibility and customer responsiveness. The subject HMLV manufacturer for this experimentation manufactures (among 8500 others) an example component for which 3 machines work independently to perform the necessary operations to manufacture this component. The experiment that is the subject of this research seeks to adapt LSS philosophies to develop treatments to improve the performance of the manufacturing of this component. These LSS-inspired treatments included 1) using cellular manufacturing methods, and the 3 machines as a single work cell to manufacture the component, and 2) using a single multipurpose machine to perform all operations required to manufacture the component. The results of this experiment demonstrate that the cellular manufacturing method was the most effective to reduce costs, to standardize operations at a process level, and to increase throughput. The single machine processing method improved production rates and on-time delivery relative to the baseline, but greatly increased lead time, thereby increasing total cost per part. These results highlight the importance of critically assessing the application of LSS within HMLV environments compared to the Low-Mix High-Volume (LMHV) environments where LSS is traditionally successful. HMLV manufacturers and researchers can use these findings to identify the most effective methods for their specific needs and to design interventions that will improve system-level manufacturing performance in high mix environments.

## 1. Introduction

The manufacturing landscape is rapidly evolving, and much of the manufacturing industry has adopted high-mix strategies to compete globally. High-mix low-volume (HMLV) manufacturers are those that produce a large variety of products and components in relatively small

2024): https://doi.org/10.5061/dryad.8pk0p2nvv
Current access is provided as URL.

**Funding:** The authors received no specific funding for this work.

**Competing interests:** I have read the journal's policy and the authors of this manuscript have the following competing interests: Amanda Normand was employed by the manufacturer while conducting this research. Thomas Bradley declares no competing interests. This does not alter our adherence to PLOS ONE policies on sharing data and materials.

quantities [1]. The HMLV environment embraces high variability in processes, demand rates, and product complexity because they allow for customization as a competitive strategy [2]. HMLV manufacturing as a category has been rapidly growing since the 1970's [3] despite global competition for low-cost production [4]. HMLV focuses on customer-driven product customization, and specialization that high volume manufacturing cannot easily adapt to [5].

Lean and Six Sigma (LSS) are typically conceptualized and executed together as a combination of existing industrial paradigms, and both have been widely applied to conventional LMHV manufacturing. Lean is a method of improving manufacturing processes through the removal of waste from the system. Six Sigma is a means of statistically controlling processes [6]. Six Sigma asserts that quality values, like feature tolerances, tended to fall on a normal distribution curve when the process was "in control". When the process requires correction, the distribution of measurements will be skewed. This insight allows manufacturers to focus on process corrections rather than constantly adjusting processes which can be expensive and unnecessary. LSS is also a departure from traditional measurements, such as defects per million, which provide in-process quality controls and allow for corrections to keep the process in control [7].

## 1.1 Literature review

The HMLV manufacturing environment presents unique challenges in the application of LSS industrial paradigms, which have traditionally been applied to great benefit in Low-Mix High-Volume (LMHV) manufacturing [8]. As summarized in Table 1, there are differences between HMLV and LMHV manufacturing that challenge the direct applicability of LSS in HMLV manufacturing environments.

In addition to the practical examples of this disconnect presented in Table 1, there are many philosophical aspects of LSS that require a re-envisioning of the context of HMLV. For example, although the fundamental Lean concept of "waste" is fundamental, the types of waste in HMLV manufacturing are different than those in LMHV environments. Definitionally, "Waste" includes transportation, inventory, motion, waiting, overproduction, over-processing, defects, and skills [6]. Lean Manufacturing focuses on reducing these types of waste in production processes [10]. In HMLV manufacturing overproduction waste is particularly relevant, because although inventory can be wasteful, inventory is also effective when used as a buffer against the varying demand patterns that are amplified by customization efforts [11]. The customization of products creates complexity in scheduling and load leveling for HMLV manufacturers. These production variations will influence motion waste, transportation waste, and will complicate the layout of work area that can accommodate the diverse value streams. This inhibits a smooth production flow and creates areas where operational "bottlenecks" occur. In LMHV manufacturing, these bottlenecks typically have easily identifiable and predictable inputs and outputs. In HMLV manufacturing, this diversity of activities and lack of repetition makes identification of bottlenecks more difficult.

In HMLV environment, waiting waste may also manifest differently than in LMHV environments. In LMHV, waiting waste is typically due to long production runs. In HMLV, the large degree of product variation means that changeovers are instead a primary source of waiting waste. Product variation can contribute to increased motion and transportation waste because of the difficulty and complexity of defining a work area layout that enables multiple converging value streams. Value streams can be defined as a map of how product flows through a production system [10]. Value Stream Mapping is typically used to identify areas of production where inefficiencies and waste occur to improve the flow and eliminate these wastes [12]. Considering this complexity and the flexibility that HMLV environments must

**Table 1. Summary of key differences between LMHV and HMLV manufacturing.**

| LMHV Manufacturing Characteristics (Traditional LSS Domain) | HMLV Manufacturing Characteristics | Challenges to LSS Philosophies in application to HMLV [9] |
|---|---|---|
| Low mix of product in high volumes | High mix of product in low volumes | LSS focuses on product standardization, as opposed to the high mixes found in HMLV |
| Economies of scale exist at a part level | Low volume for all parts | LSS's emphasis on economies of scale exist only at process-level in HMLV |
| Low customization of product | High customization of product | LSS focuses on reducing production variation through standardized product while HMLV uses product customization as a sales strategy. |
| More likely offshore manufacturing (U.S.) | More likely onshore manufacturing (U.S.) | LSS reduces the labor skillset needed through standardization while HMLV relies on highly skilled labor. |
| Volume-based cost strategy | Customization-based cost strategy | LSS takes advantage of economies of scale with waste reductions. HMLV environments have significant process variability that limits the impact of small improvements over time at a product level. |
| Typically designed for unskilled labor | Typically designed for skilled labor | LSS reduces product variation which reduces the labor skill level needed. Product variation inherent to HMLV environments requires flexibility and adaptability in the workforce skillsets. |

maintain, we can also understand that the data-driven approach inherent to LSS, must also be adapted to smaller batches and to more diversity in order to be successful in waste reduction. When employing LSS philosophies and techniques, adaptations to the specific environment and context of the process to be improved is necessary [13]. This same re-envisioning of the precepts of LSS is applicable to many of the challenges (as in Table 1) of applying LSS to HMLV.

Based on this understanding of the applicability and value of LSS in HMLV environments, we can identify that there is a need to measure the efficacy of these types of adaptations of LSS in HMLV practice. This research therefore presents a set of adaptations of LSS philosophies, metrics, and interventions to meet the needs of a HMLV manufacturer located in Wisconsin, USA. We present an experimental evaluation and assessment of these interventions and discuss the implications of these findings to the more general question of the applicability of LSS to the HMLV manufacturing environment. Conclusions focus on the definition of specific LSS philosophies that can be used to inspire improvements in HMLV manufacturing.

## 2. Methods

This section presents the methods by which we define, measure and test a set of LSS-inspired interventions in a HMLV manufacturing environment. Following LSS philosophies, the Define, Measure, Analyze, Improve, Control (DMAIC) process was used for each experimental intervention. This is further detailed for each intervention in S1 Appendix.

### 2.1. HMLV manufacturing site

A set of experiments was conducted in practice within an operating HMLV manufacturing environment to assess how the philosophies of LSS could be effectively applied and evaluated. The location of these experiments was an operating HMLV manufacturing plant in Wisconsin, USA. This plant is classifiable as a HMLV environment in that it manufactures 8,500-part numbers annually with an average batch quantity of 40 components.

Within this HMLV environment, there are several "work centers" that are part of the many converging value streams. Each work center is used as needed based on the production demand that is both volatile and continuously changing. This aspect of HMLV manufacturing allows for experimentation without significant production disruption if conducted while a work center has lower volume production flow.

## 2.2. Adaptation of LSS to HMLV

LSS philosophies include many potential benefits for HMLV manufacturers that could help resolve some of the major detractors from competitive advantage. These philosophies provide a framework that aims to reduce waste and enhance process flow through reduced process variation and defects. This aligns closely with the objectives of HMLV manufacturers that seek to improve resource utilization, minimize costs, and smooth production flow while maintaining the flexibility required to provide highly customized products. It is hypothesized that improving flow in this production environment will improve the overall cost to manufacture components.

These experiments specifically targeted process-level production streamlining instead of product-level interventions. By focusing on process-level improvements, these experiments aimed to maximize standardization within the HMLV environment, leading to enhanced waste reduction outcomes.

The specific HMLV manufacturer where this study was conducted had multiple CNC turning centers. The number of operators available was less than the number of turning centers available. In LMHV manufacturing, long runs (quantity) of components enables operators to run multiple machines where they focus on loading and unloading materials to keep production continuous. Tooling change-overs between components are infrequent and, depending on volume, may be unnecessary [14]. To do this, highly specialized and automated equipment is used that is typically dedicated to a specific product or product line. This dedicated equipment is a large capital investment that seeks to improve productivity but reduces flexibility.

The need for flexibility in HMLV manufacturing can have a direct negative impact on equipment uptime because it adds to operational complexity. To improve equipment uptime, specialized equipment and techniques are used. However, with the large variety of components, this becomes impractical from a capital investment standpoint. The potential answer to this is to focus on less specialized operations that can be used for multiple components by reducing them to their basic functions and creating efficiency at that level.

## 2.3. Baseline manufacturing operations

At a process level, we can understand that many unrelated components have similar processing steps, requiring coordination and consideration in the HMLV environment. As illustrated in Fig 1, mapping these process steps for multiple components shows crossing and overlapping paths for product flow [15].

A single part number (Component A from Fig 1) was chosen for experimentation. As shown in Fig 2, the baseline operations to complete this component included turning, grinding, and hobbing.

## 2.4. Experimental interventions

For this experiment, both highly specialized manufacturing processes that are refined for the specific component, and more basic manufacturing techniques that are refined at the process level were compared. Consideration was also given to the volume and frequency of manufacturing where this component has relatively higher volume and frequency than other components and provided the best opportunity for experimentation without creating unnecessary production. Before the experiment, the component was manufactured in multiple operations that were used in many value streams. The scheduling of multiple value streams created WIP between operations. This WIP, as part of the overall cost to manufacture, was addressed in the experiment using both single machine manufacturing and cellular manufacturing.

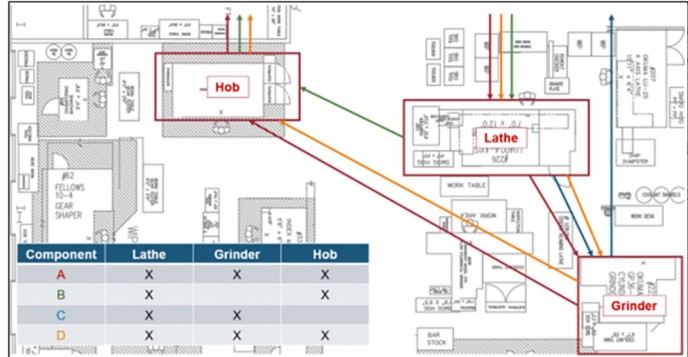

**Fig 1. Process flow for (4) unrelated components in a HMLV manufacturing environment.**

The production operations were examined for opportunities for standardization that allowed for maintained operational flexibility. The opportunity was identified at a process level where multiple similar components could be grouped by their shared processing methods.

Two different interventions were chosen based on the LSS philosophy of waste reduction:

**2.4.1. Intervention 1: Cellular manufacturing.** A work cell was created that included the processes required to manufacture the component(s). As shown in Fig 3, this work cell allowed for the continuous flow of components without wait time between processing operations. This processing method also allowed functions that would be considered "waste" in LSS to become internal operations. For example, setup time for each machine, part changeovers, and inspections between operations for quality control can now be considered internal to the cellular manufacturing system. Under this intervention, the operators loaded the raw materials for the first operation (the lathe) and then moved them to the remaining operations (grinder and hob) for processing. The cell produced completed components with the 3 separate machines.

**2.4.2. Intervention 2: Single machine processing.** The second intervention involved the use of a single machine to process the component(s) completely. For this, a lathe with live tooling was chosen. This lathe was capable of turning the components, including a turning operation capable of the same surface finish as grinding, and cutting the spline teeth with a single tooth cutter rather than a hob. As shown in Fig 4, the operator only needed to load the raw material and then unload the completed component(s).

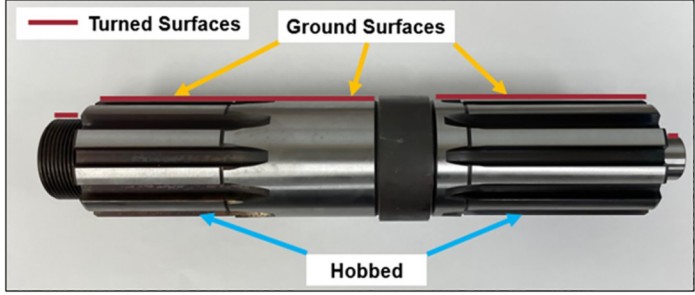

**Fig 2. Shaft used for experimentation of processing method adaptations for process level standardization.**

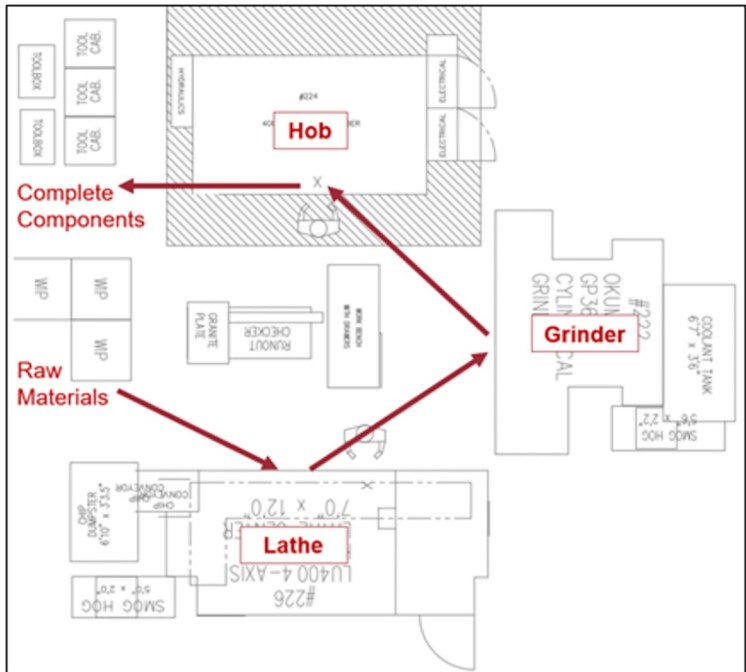

**Fig 3. Process flow for cellular manufacturing intervention.**

## 2.5. Metrics

To effectively assess the impact of the application of the principles of LSS, a set of metrics and specific measurement methods, including the variables (Table 2), were defined.

1. Work In Process (WIP) = IW × OH$_{EX}$

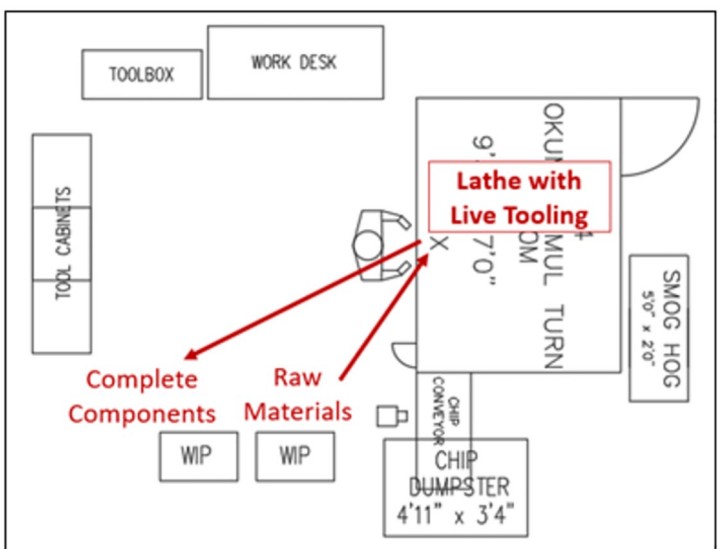

**Fig 4. Process flow for single machine processing.**

**Table 2. Variables measured to calculate the performance metrics.**

| Variable | Definition |
|---|---|
| $C_S$ | Cost of space (USD/ft$^2$) |
| D | Defects (Number of Components as a percentage) |
| IW | Inventory value in WIP (USD) |
| $M_S$ | Machine Setup Time (hrs) |
| $OH_{EX}$ | Overhead multiplier for the experiment period (time) |
| $OH_{WC}$ | Overhead for the parts produced (3 months) (USD) |
| $OW_{WC}$ | Operator wage (USD/ hr) |
| $Q_B$ | Quantity of components in batch (Qty) |
| $Q_S$ | Quantity of components scrapped in batch (Qty) |
| $\sigma$ | Standard Deviation, - |
| $T_C$ | Cycle Time (per part) (hrs) |
| $T_J$ | Time operator is on specific job (hrs) |
| $T_O$ | Labor time for operator (hrs) |
| $T_P$ | Planned production time (hrs) |
| $\bar{x}$ | Mean or average change in process over time (hrs) |

2. On Time Delivery (OTD) = Quantity Batches Completed on Time/ Total Batches

3. Performance ($H_C$) = $T_C$ / $Q_B$

4. Effectiveness (E) = $(((Q_B - Q_S) \times T_P) / T_O) / 100$

5. Uptime ($H_U$) = $(T_C \times Q_B) / T_J$

6. Lead Time (LT) = Average Hours Per Batch

7. Process Variability (Cpk) = $\bar{x}/\sigma$

8. Parts Per Hour (PPH) = $Q_B / T_J$

9. Total Cost ($C_P$) = $(T_C + (M_S / Q_B)) \times (OW_{WC} + OH_{WC})$

Each of these variables and metrics were measured and calculated for the duration of the experimental interventions. The baseline period and each of the interventions were implemented in a 3-month period (each, totaling 9 months). Operators were instructed to run the components as normal, where each machine was part of the larger mixed value stream, for the baseline period. For the cellular intervention, operators were instructed to use the work cell as a single entity where single piece flow was achieved for the batch of components. For the single machine processing intervention, the operator was instructed to run the entire batch on a single machine.

For both the baseline and the experimental periods, production demand was typical, and all machines were expected to perform normal production operations, including other components, during these periods.

## 3. Results

This section presents the results of the baseline and two interventions. A summary of the results allows for direct comparison of each of the manufacturing setups along multiple dimensions of LSS performance.

| | | 12.23 | | | 8.63 | | | 16.57 | 37.44 | Total Uptime (H₍ᵤ₎) |
|---|---|---|---|---|---|---|---|---|---|---|

(Figure 5 table image)

**Fig 5. Value stream map of baseline processing with machines working separately in mixed value streams (all values in hours).**

In HMLV environments, traditional parametric statistical tools cannot be defensibly applied due to their inherent assumptions of large sample sizes and stable data distributions [16]. In HMLV environments, data is sparse and non-stationary (see S2 Appendix for details), rendering these assumptions invalid. All data presented in this study are presented as values that are the integrated result over the duration of these experiments. This provides more accurate and comprehensive understanding of the HMLV performance in practice, including variability and non-stationarity of the system under test. This approach recognizes the dynamic nature of HMLV environments and provides analytical methods to suit.

### 3.1. Baseline manufacturing system performance

For the 3 months prior to implementation of any of the interventions, the performance of the baseline manufacturing system was determined by measuring each of these metrics before any experimental treatment was applied. The baseline included 3 separate work centers (Lathe, Grinder, Hob) operating independently of each other. In Fig 5, a Value Stream representation of this baseline is presented. This baseline batch manufacturing process involved the three manufacturing processes (Lathe, Grinder, Hob), with staging and setup before each of these. During the baseline measurement period, Part A was manufactured 58 times (1 complete batch), and Fig 5 presents the summation of those parts' results.

With the work centers acting as separate entities and as part of a larger system of mixed value streams, WIP is present at each machine waiting to be processed. Batch processing dictates that a batch of components is complete before any single component is considered complete and logged into stock. As shown in Fig 1, this means that a new job or work packet will be in queue for an average of 67.32 hours before it begins to process in the first operation (Lathe). The critical path for this processing method, for a batch to be completed, was 253.61 hours. The uptime for this method is measured at 14.76% of the total processing time with wait time in the multiple staging events providing the most significant portion of downtime. The entire set of metrics of performance are summarized in Table 3.

### 3.2. Manufacturing system performance under cellular manufacturing intervention

Intervention #1 used the same machines as the baseline but now configured as a cellular workflow where the queue for work existing in front of the first operation and then components flowed through the work cell to the second and third operations, as represented in the Value Stream Map shown in Fig 6.

With the work centers acting as a cellular entity, WIP is reduced to a single location ahead of the cell. The time to set up the next operation was also internalized where it was completed during the previous operation. The reduction in wait time and internalizing of set up time for the grinder and hob operations reduced the total processing time. As shown in Fig 5, the total wait time in staging was reduced to 67.32 hours and the set-up time, although not reduced overall, only contributed to 4.75 hours of actual down time for the cell. The critical path for

**Table 3. Summary of measurements for each processing method used.**

| Variable | Definition | Baseline | Cellular | Single WC |
|---|---|---|---|---|
| $C_S$ | Cost of space | $112,500.00 | $60,000.00 | $37,500.00 |
| D | Defects | 0.12% | 0.10% | 0.26% |
| IW | Inventory value in WIP | $73,193.99 | $11,820.80 | $44,609.36 |
| $M_S$ | Machine Setup Time | 3.38 | 2.84 | 6.11 |
| $OH_{EX}$ | Overhead multiplier for the experiment period | 0.10 | 0.10 | 0.10 |
| $OH_{WC}$ | Overhead for the parts produced (3 months) | $77.23 | $77.23 | $77.23 |
| $OW_{WC}$ | Operator wage | $26.25 | $26.25 | $26.25 |
| $Q_B$ | Quantity of components in batch | 58 | 58 | 58 |
| $Q_S$ | Quantity of components scrapped in batch | 0.07 | 0.06 | 0.15 |
| σ | Standard Deviation | 851.14 | 361.73 | 412.02 |
| $T_C$ | Cycle Time (per part) | 0.59 | 0.59 | 1.63 |
| $T_J$ | Time operator is on specific job | 11.97 | 9.85 | 47.49 |
| $T_O$ | Labor time for operator | 11.97 | 9.85 | 29.34 |
| $T_P$ | Planned production time | 9.86 | 8.14 | 9.44 |
| X-bar | Mean or average change in process over time | 768.83 | 548.84 | 1047.87 |

this processing method, for the batch to be completed, was 109.51 hours. The uptime for this method is measured at 34.19% of the total processing time with wait time and down time reduced. The entire set of metrics of performance are summarized in Table 3.

### 3.3. Manufacturing system performance under single machine processing intervention

Intervention #2 used a single machine to manufacture the component in a single setup. There were many considerations to do this that required some changes in processing methods to machine the component to the same specifications as with multiple machines. For instance, plunge grinding was not an option inside of the lathe without additional machine modification and reduction in machine and tool life due to abrasives used, and therefore increased the processing time to turn the required surface finish. Hob operations were performed using a single tooth cutter in live tooling.

As shown in Fig 7, the wait time in staging was more than the wait time in the cellular processing method but less than the total wait time for the baseline processing method. The reduction in total wait time in staging, compared to the baseline, increased the uptime compared to the baseline. However, the processing time for the lathe to perform the 3 required operations was increased. The critical path for this processing method was 210.13 hours to complete the component.

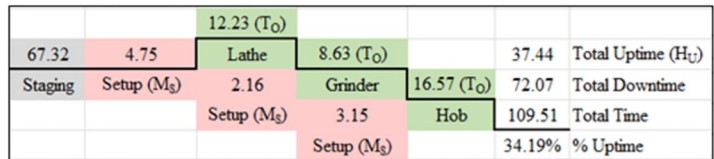

**Fig 6. Value stream map of cellular workflow processing with machines working as a single work cell (all values in hours).**

| | | 39.44 | 39.44 | Total Uptime ($H_U$) |
|---|---|---|---|---|
| 162.64 | 8.05 | Lathe | 170.69 | Total Downtime |
| Staging | Setup ($M_S$) | | 210.13 | Total Time |
| | | | 18.77% | % Uptime |

**Fig 7. Value stream map of single machine processing (all values in hours).**

The entire set of metrics of performance are summarized in Table 3.

## 3.4. Results summary and synthesis

The measured variables for the baseline and both experimental methods can be seen in Table 3. As shown, the space required for single machine processing was less than the cellular processing method. When compared to the baseline, the cellular processing method required less space because of the reduction in staging space needed and the ability to overlap operator work envelopes between machines.

Using these variables, the LSS-informed metrics of performance were calculated as presented in Table 4.

This experiment provides evidence of the benefits available from both single machine and cellular processing interventions in HMLV manufacturing. Both interventions provided improvements in cost, parts per hour, and WIP compared to the baseline. Cellular processing provided a larger benefit in each category in addition to improved overall lead time and quality. For single machine processing, the added cycle time to perform all of the operations negatively impacted the performance of Intervention 2 by these metrics compared to cellular processing.

During the experimental period, there were also extraneous factors that affect the replicability and applicability of these measurements. The main impact came from global supply chain challenges due to the COVID-19 pandemic, which disrupted the material availability and changed the lead times resulting in some work packets completed well in advance of their due date and others were rushed through once materials were available. The results presented here are asserted to be replicable and applicable, other experiments performed in 2020 are not presented here due to confounding with the COVID-19 pandemic.

**Table 4. Calculated metrics for the 3 processing methods for the component.**

| Eq # | Metric | Variable | Units | Baseline | Cellular | Single WC |
|---|---|---|---|---|---|---|
| 1 | WIP | **WIP** | USD ($) | $7,624.37 | $1,231.33 | $4,646.81 |
| 2 | On Time Delivery | **OTD** | % | 33.33% | 33.33% | 85.71% |
| 3 | Performance | $\mathbf{H_C}$ | Hours | 0.36 | 0.36 | 0.40 |
| 4 | Effectiveness | **E** | % | 47.68% | 47.74% | 18.65% |
| 5 | Uptime | $\mathbf{H_U}$ | Hours | 14.76% | 34.19% | 18.77% |
| 6 | Lead Time | **LT** | Hours | 768.83 | 548.84 | 1047.87 |
| 7 | Process Variability | **Cpk** | Cpk | 1.21 | 1.38 | 0.76 |
| 8 | Parts Per Hour | **PPH** | Parts | 0.19 | 0.79 | 0.83 |
| 9 | Total Cost (per part) | $\mathbf{C_P}$ | USD ($) | $66.69 | $64.81 | $76.00 |

## 4. Discussion

### 4.1. The application of Lean Six Sigma to high-mix low-volume manufacturing

If we consider results of these experiments in the context of conventional LSS philosophies, the baseline manufacturing configuration might be considered wasteful and inefficient. The baseline is measured as having significant added costs due to WIP ($7,624.37 compared to $1,231.33 for cellular and $4,646.81 for Single WC) between operations. This resulted in a relatively long wait time, and large overall time required to produce the components. Both the Cellular Manufacturing method, and the Single Machine method would be considered promising LSS interventions in a LMHV manufacturing environment, relative to the baseline, because of their potential to reduce WIP, reduce setup time ($M_S$), and thereby reduce waste.

The results of these experiments in a HMLV manufacturing environment instead illustrate that these tradeoffs are more complicated than might be conventionally considered. For the Single Machine Manufacturing method, the component was manufactured with significantly reduced process intervention (an operator wasn't moving components), but in this HMLV application, the results of this experiment show that there is a reduction in product quality (Cpk) that overwhelms the benefits from reduced operator intervention. In this HMLV application, the types of parts that must be manufactured with this Single Machine are so numerous, that the multi-step manufacturing process is difficult to control. The complexity of machine setup ($M_S$), of inter-machining-step quality control, and of labor meant that the Single Machine manufacturing method produced lower quality parts ($Q_S$) that had to be reworked to meet specifications. The Single Machine processing method also reduced the cost of WIP, but decreased process effectiveness (E) due to quality problems that were the result of the increased complexity of machine set up.

On the other hand, for the Cellular Manufacturing method, the experimental evidence indicates that internalizing non-valued added activities (i.e., "waste") into value-added resulted in decreased production time ($T_C * Q_S + M_S$) and fewer quality errors ($Q_S$). In this HMLV environment, manufacturing quality was improved because of the frequent human interventions and in-process quality checks. The Cellular Manufacturing method also significantly reduced the cost of WIP because of its increased throughput and reduced wait time.

These findings illustrate that although the philosophies and concepts of LSS are fundamental to improving productivity, the unique demands of the HMLV environment mean that many of the conventional LSS metrics and concepts that have been successfully applied to LMHV manufacturing must be re-validated in application to HMLV manufacturing.

### 4.2. Implications for the applicability of single machine processing

Single Machine Processing is often presented in literature as an ideal case in which to realize LMHV manufacturing quality because it allows for higher accuracy between features by removing the need to control interactions between machines [17]. Instead, as highlighted in Tables 3 and 4 in this HMLV experiment, the Single Machine intervention had the lowest process quality level (E $_=$ 18.65% compared to 47.68% and 47.74% respectively). The Single Work Center method had measurably lower process control (ie. lower Cpk) than the other methods.

In the HMLV manufacturing environment, these quality problems were largely the results of the increased complexity of the machine setup, and of fewer opportunities to measure and adjust the machine during processing. The resulting quality issues negated the Single Work Center's improvement in parts per hour compared to the other processing methods. These results point to the importance of very strong quality controls for the Single Work Center

method. If stronger quality controls during setup and during processing had been realized, the Single Work Center method might have been able to realize an increase in production rate (0.83 PPH, compared to 0.19 for the baseline and 0.36 for the cellular intervention). In this experiment, and in this HMLV application, the highly specialized equipment that would be required to accommodate the high numbers of different components, and the high volatility associated with very small volumes, was cost-prohibitive.

We also observed that skill level required of the operator for the Single Machine method is higher than the other manufacturing systems studied here. The operator needed to be capable of setting up more than one type of machine operation and needs to do so in a machine that was more complex than those used in the baseline and the cellular interventions. In the cellular intervention, there was also an increase in the required skill level because there was a requirement that each of the operators were capable of at least (2) different machine operations. The requirement of highly skilled labor is a typical constraint in HMLV manufacturing. Although it increases workforce flexibility in terms of labor skillsets, it may reduce flexibility in terms of change management [18].

## 4.3. Implications for the applicability of cellular manufacturing

Cellular Manufacturing in a HMLV environment requires the development of groups of processes that can be executed together in a manufacturing cell [15]. Cellular manufacturing is also more difficult to set up in HMLV environments where the required equipment is not portable and is not reconfigurable at the same rate that the product changes. In HMLV manufacturing, production demand patterns frequently change, leading to processing methods that are poorly compatible with existing work cells.

Instead, in HMLV manufacturing, manufacturing cells must be constructed to serve commonly applied sets of operations, which would apply to a wide variety of products and product families. Using the workflow mapping technique illustrated in Fig 1, we developed an understanding of commonalities in the product which allow for common processing methods. If all the components were mapped in this environment, stronger trends would be apparent that would potentially allow for additional manufacturing cells to be created to achieve the same successes.

Although the Cellular Manufacturing work cell required more space in the manufacturing facility compared to the single machine processing, there was a significant advantage in the cost of WIP in the work cell (approximately 84% less than the baseline and 74% less than the single machine intervention). This was the result of faster processing with setups internal to cycle times, and single piece flow through the work center. These findings are consistent with the benefits that others have achieved with cellular processing methods [19].

The Cellular Manufacturing methods measurably improved the workflow and reduced quality errors compared to the baseline and the Single Machine method. The time to set up the second and third machining operations was able to be done internally to the cycle time of the previous operation which, in addition to one-piece-flow for components, reduced the critical path. This method provided an additional benefit by only requiring 1 machine operator to run all 3 machines. Quality also improved (0.06 scrap rate) compared to the baseline (0.07 scrap rate) because of the operator's ability to impact all machining operations as necessary to improve and optimize operations in sequence. The single piece-flow also decreased the wait time between operations because batches were completed through all operations using single piece flow. WIP existed only at the beginning of the value stream where it was waiting to be processed in the work cell. This reduction in wait time is consistent with removal of waste as defined philosophically by LSS [20].

## 5. Conclusions

HMLV manufacturing is an important component of the US manufacturing sector, but the philosophies and practicalities of applying LSS manufacturing paradigms to HMLV environments are less developed.

Through a set of on-site in-practice experiments with the application of two LSS-inspired interventions (Cellular Manufacturing, and Single Machine Manufacturing) to the baseline production processes at an example US manufacturer, this study has been able to quantify these interventions' costs and benefits in the HMLV environment. The experimental results of this study provide evidence that the cellular processing method resulted in the most benefits to the manufacturing environment. The cellular method resulted in less inventory value in WIP ($C_P$ = \$1,231.33 compared to \$7,624.37 for the baseline and \$4,646.81 for the Single WC), stronger On Time Delivery (OTD = 85.71% compared to 33.33% for the both the baseline and the Single WC), and the lowest total cost per part ($C_P$ = \$64.81 compared to \$66.69 for the baseline and \$76.00 for the Single WC). Together these results illustrate that cellular manufacturing method proved to be the most effective in reducing costs and improving flow, while the single machine processing method was ineffective in this HMLV manufacturing environment without further quality control measures. These results and discussion provide insights for HMLV manufacturers looking to optimize their operations through standardization at a process level that allows them to maintain operation flexibility while reducing component costs.

## Supporting information

**S1 Appendix. Appendix A: DMAIC for experimental approaches.**
(DOCX)

**S2 Appendix. Appendix B: Non-stationarity of production statistics under HMLV manufacturing.**
(DOCX)

**S3 Appendix. Appendix C: Variables and dataset.**
(DOCX)

**S1 Dataset.**
(PDF)

## Acknowledgments

This research was supported by a team of Engineers and CNC Programmers including: Shane Sullivan, Tim Dewitz, Jeff Smith, Garret Yohnk, Carl Krumenauer, Shawn Mykytiuk, Ryan Elliott, Josh Adrian, and Taylor Thompson.

## Author Contributions

**Conceptualization:** Amanda Normand, T. H. Bradley.

**Formal analysis:** Amanda Normand.

**Investigation:** Amanda Normand.

**Methodology:** Amanda Normand.

**Project administration:** Amanda Normand.

**Supervision:** Amanda Normand, T. H. Bradley.

**Visualization:** Amanda Normand, T. H. Bradley.

**Writing – original draft:** Amanda Normand.

**Writing – review & editing:** T. H. Bradley.

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
