## [Decision Letter · Decision Letter 0]

4 Dec 2023

PONE-D-23-20942An experimental investigation of Lean Six Sigma philosophies in a high-mix low-volume manufacturing environmentPLOS ONE

Dear Dr. Normand,

Thank you for submitting your manuscript to PLOS ONE. After careful consideration, we feel that it has merit but does not fully meet PLOS ONE’s publication criteria as it currently stands. Therefore, we invite you to submit a revised version of the manuscript that addresses the points raised during the review process.

**ACADEMIC EDITOR: A major revision, considering all the review comments is required.**==============================

We look forward to receiving your revised manuscript.

Kind regards,

Agbotiname Lucky Imoize

Academic Editor

PLOS ONE

Journal Requirements:

"I have read the journal's policy and the authors of this manuscript have the following competing interests: Amanda Normand was employed by the manufacturer while conducting this research.

Thomas Bradley declares no competing interests."

Additional Editor Comments:

The authors are required to revise the manuscript according to the review reports.

Reviewers' comments:

Reviewer's Responses to Questions

**Comments to the Author**

1. Is the manuscript technically sound, and do the data support the conclusions?

Reviewer #1: Partly

Reviewer #2: Partly

2. Has the statistical analysis been performed appropriately and rigorously? 

Reviewer #1: No

Reviewer #2: No

3. Have the authors made all data underlying the findings in their manuscript fully available?

Reviewer #1: No

Reviewer #2: Yes

4. Is the manuscript presented in an intelligible fashion and written in standard English?

Reviewer #1: Yes

Reviewer #2: No

5. Review Comments to the Author

Reviewer #1: This article discusses the impact of three different LSS interventions on HMLV manufacturing. The article is well within the scope of the Journal. However, some shortcomings regarding the text, literature review, sufficient experimental details, and discussion need to be addressed before publication. These are some suggestions to improve the article's quality.

1. The article lacks a comprehensive literature review on LSS and HMLV manufacturing. It would be beneficial to add a dedicated section for this purpose.

2. Experiments data presented is insufficient to conclude the results.

3. The inclusion of better pictorial representations of the experiments would enhance reader engagement.

4. While LSS philosophies typically follow the DMAIC approach, this article does not delve into this aspect. A significant revision is required to improve the overall structure.

5. Review the sentences in line number 24 and line number 73 for accuracy and clarity.

6. The authors make use of VSM, but they have not provided an explanation of what it is or why it's being used.

7. The concept of staging events and their impact on process downtime is not adequately explained. This should be addressed for a more comprehensive understanding.

8. To enhance the quality of this article, it is recommended to refer to the following related sources: https://doi.org/10.1016/j.heliyon.2022.e09043, https://doi.org/10.1108/IJQRM-08-2022-0250, https://doi.org/10.1080/09537287.2016.1185188

9. The outcomes of the study are not presented in the conclusion section. Specific conclusions are required.

Reviewer #2: The presented article has potential, but in my view, it has some issues. The structure of the work is not adequate: i) the introduction is long and should not contain figures or tables. The research question, objectives, and hypotheses should be evident. Part of what is in the review should go to a literature review chapter; ii) the method is not clear. Statistical hypotheses are not defined, and independent or causal variables are not identified. The experiment design is not clearly explained. The estimates of the indicators are punctual. It would be more appropriate to consider the confidence intervals and statistical significance. For example, capability indices. iii) The tables are presented far from the text, in an unordered way, and are not clear. iv) The analyses should be based on the variables identified in the method section. v) The conclusions should adhere to the objectives and statistical hypotheses.

6. PLOS authors have the option to publish the peer review history of their article (what does this mean?). If published, this will include your full peer review and any attached files.

Reviewer #1: No

Reviewer #2: No

---

## [Author Response · Author response to Decision Letter 0]

22 Jan 2024

Reviewer #1 Comment Responses

Reviewer #1: This article discusses the impact of three different LSS interventions on HMLV manufacturing. The article is well within the scope of the Journal. However, some shortcomings regarding the text, literature review, sufficient experimental details, and discussion need to be addressed before publication. These are some suggestions to improve the article's quality.

1. The article lacks a comprehensive literature review on LSS and HMLV manufacturing. It would be beneficial to add a dedicated section for this purpose.

Author Response: Thank you for this suggestion.

Author Action: We have added a new section after the introduction titled “Literature Review” and revised the content (1.1.). Please find this section highlighted in green in the “Revised Manuscript with Track Changes”.

2. Experiments data presented is insufficient to conclude the results.

Author Response: Thank you for your comment.

Author Action: Additional data has been made available under https://doi.org/10.5061/dryad.8pk0p2nvv . 

Data will be available there after our libraries’ internal checks and assurances are completed. 

Bradley, Thomas; Normand, Amanda (Forthcoming 2024). An experimental investigation of Lean Six Sigma philosophies in a high-mix low-volume manufacturing environment [Dataset]. Dryad. 

3. The inclusion of better pictorial representations of the experiments would enhance reader engagement.

Author Response: We agree that additional pictorial representations would enhance reader engagement. 

Author Action: We have added an additional figure (Figure 4) to better describe the second experimental intervention (single machine processes) to share the same type of pictorial representations as the baseline and intervention 1.

4. While LSS philosophies typically follow the DMAIC approach, this article does not delve into this aspect. A significant revision is required to improve the overall structure.

Author Response: Thank you for your comment. This paper is concentrating more on the use of Lean Six Sigma philosophies to improve high-mix low-volume manufacturing environment. We concentrate in this paper on comparing Lean Six Sigma to other methods. Please note that this paper is a set of experiments rather than a case study.

Author Action: Appendix A has been added to discuss the characteristics of the DMAIC approach for this experiment.

5. Review the sentences in line number 24 and line number 73 for accuracy and clarity.

Author Response: Thank you for your careful examination of the manuscript.

Author Action: Lines 24 and 73 were updated for accuracy and clarity.

6. The authors make use of VSM, but they have not provided an explanation of what it is or why it's being used.

Author Response: We agree that a stronger explanation of what a VSM is and why it is used is needed.

Author Action A definition, including the intended use of VSM’s has been added in Section 1.1. Please find these changes highlighted in green in the “Revised Manuscript with Track Changes” document. 

7. The concept of staging events and their impact on process downtime is not adequately explained. This should be addressed for a more comprehensive understanding.

Author Response: Thank you for this insight.

Author Action: A description of the unique characteristics of HMLV manufacturing that allow for experimentation without significant production disruption has been added to section 2.1. Please find this update highlighted in green in the “Revised Manuscript with Track Changes”.

8. To enhance the quality of this article, it is recommended to refer to the following related sources: https://doi.org/10.1016/j.heliyon.2022.e09043, https://doi.org/10.1108/IJQRM-08-2022-0250, https://doi.org/10.1080/09537287.2016.1185188

Author Response: Thank you for your suggestions, we agree that these referenced articles will enhance the quality of the article.

Author Action: These article references have been added to the manuscript in Section 1.1. Please find these additions highlighted in green in the “Revised Manuscript with Track Changes” document.

9. The outcomes of the study are not presented in the conclusion section. Specific conclusions are required.

Author Response: Thank you for this comment, the article has been revised to highlight the conclusions more explicitly. 

Author Action: Updates have been made to the conclusions section to better describe the variables, metrics, and their values within the discussion. Please see these changes highlighted in green in the “Revised Manuscript with Track Changes”. 

Reviewer #2 Comment Responses

Reviewer #2: The presented article has potential, but in my view, it has some issues. The structure of the work is not adequate: 

i) the introduction is long and should not contain figures or tables. The research question, objectives, and hypotheses should be evident. Part of what is in the review should go to a literature review chapter

Author Response: We agree that a dedicated Literature Review section would be beneficial as well as more explicit statements of the research questions, objectives, and hypotheses. 

Author Action: A dedicated Literature Review section has been added and significant changesd made to provide clarity. Please find these changes highlighted in green in the “Revised Manuscript with Track Changes”.

ii) the method is not clear. Statistical hypotheses are not defined, and independent or causal variables are not identified. The experiment design is not clearly explained. The estimates of the indicators are punctual. It would be more appropriate to consider the confidence intervals and statistical significance. For example, capability indices. 

Author Response: Thank you for your comment. We recognize that the use of statistical tools is difficult in HMLV environments given the inherent variability and unstable data distributions. 

Author Action: A description of the processes for statistical analysis (n=1) and an understanding of why this approach is necessary have been added to the conclusions sectionSection 3, footnote 1, and Appendix B has been added for additional detail. Please find these updates highlighted in green in the “Revised Manuscript with Track Changes”. 

iii) The tables are presented far from the text, in an unordered way, and are not clear. 

Author Response: We agree that this is a formatting challenge.

Author Action: We will support and provide feedback to the typesetting staff when it is typeset for final publication.

iv) The analyses should be based on the variables identified in the method section. 

Author Response: Thank you for your comment. We agree this would further enhance the article.

Author Action: Values, variables, and metrics have been more explicitly identified within the text of the document to refer to those listed in the methods section. Please see these changes highlighted in green throughout the text in the “Revised Manuscript with Track Changes”.

v) The conclusions should adhere to the objectives and statistical hypotheses.

Author Response: We agree that a better description of the processes and, more specificallyspecifically, the reasoning for the processes used would be helpful.

Author Action: Several updates have been made, which are highlighted in green in the “Revised Manuscript with Track Changes”. A significant change to the description of the processes used has been added to Section 5.

---

## [Decision Letter · Decision Letter 1]

12 Feb 2024

An experimental investigation of Lean Six Sigma philosophies in a high-mix low-volume manufacturing environment

PONE-D-23-20942R1

Dear Dr. Normand,

We’re pleased to inform you that your manuscript has been judged scientifically suitable for publication and will be formally accepted for publication once it meets all outstanding technical requirements.

Kind regards,

Agbotiname Lucky Imoize

Academic Editor

PLOS ONE

Additional Editor Comments (optional):

Accept in current form.

Reviewers' comments:

Reviewer's Responses to Questions

**Comments to the Author**

1. If the authors have adequately addressed your comments raised in a previous round of review and you feel that this manuscript is now acceptable for publication, you may indicate that here to bypass the “Comments to the Author” section, enter your conflict of interest statement in the “Confidential to Editor” section, and submit your "Accept" recommendation.

Reviewer #1: All comments have been addressed

Reviewer #2: All comments have been addressed

2. Is the manuscript technically sound, and do the data support the conclusions?

Reviewer #1: Yes

Reviewer #2: Yes

3. Has the statistical analysis been performed appropriately and rigorously? 

Reviewer #1: Yes

Reviewer #2: Yes

4. Have the authors made all data underlying the findings in their manuscript fully available?

Reviewer #1: Yes

Reviewer #2: Yes

5. Is the manuscript presented in an intelligible fashion and written in standard English?

Reviewer #1: Yes

Reviewer #2: Yes

6. Review Comments to the Author

Reviewer #1: The paper may be accepted for publication

Reviewer #2: The paper has been improved, and the reseach issue is very important: the application of Lean Six Sigma (LSS) principles in a High-Mix Low-Volume (HMLV) manufacturing setting. The introduction is shorter than in the first version. Everything has been improved.

7. PLOS authors have the option to publish the peer review history of their article (what does this mean?). If published, this will include your full peer review and any attached files.

Reviewer #1: No

Reviewer #2: **Yes: **Pedro Carlos Oprime
